# DiverseAgentEntropy: Quantifying Black-Box LLM Uncertainty through Diverse Perspectives and Multi-Agent Interaction

## Abstract

Quantifying the uncertainty in the factual parametric knowledge of Large Language Models (LLMs), especially in a black-box setting, poses a significant challenge. Existing methods, which gauge a model's uncertainty through evaluating self-consistency in responses to the original query, do not always capture true uncertainty. Models might respond consistently to the origin query with a wrong answer, yet respond correctly to varied questions from different perspectives about the same query, and vice versa. In this paper, we propose a novel method, DiverseAgentEntropy, for evaluating a model's uncertainty using multi-agent interaction under the assumption that if a model is certain, it should consistently recall the answer to the original query across a diverse collection of questions about the same original query. We further implement an abstention policy to withhold responses when uncertainty is high. Our method offers a more accurate prediction of the model's reliability and further detects hallucinations, outperforming other self-consistency-based methods. Additionally, it demonstrates that existing models often fail to consistently retrieve the correct answer to the same query under diverse varied questions even when knowing the correct answer.

## 1 Introduction

Large language models(LLMs) demonstrate impressive capabilities in encoding real-world knowledge within their parameters and utilizing this knowledge to support knowledge-intensive tasks (Yu et al., 2024). However, these systems may resort to hallucinations (Ji et al., 2023) when the necessary knowledge is missing, unreliable, inaccurately stored, or not retrieved even if it exists within the model's parametric knowledge. In the future, to build and deploy powerful AI responsibly, we will need to develop robust techniques for scalable oversight (Bowman et al., 2022): alignment methods that scale with a model's capabilities. When models become increasingly powerful but still suffer from hallucinations (Nananukul & Kejriwal, 2024), users must find ways to identify and extract trustworthy knowledge from these untrustworthy models. Since most users interact with LLMs via API calls (Anthropic, 2024; OpenAI et al., 2024), we focus on the black-box model setting, ensuring that our solution applies to any model without requiring internal access to weights or gradients, or external assistance such as expert consultation or retrieval augmentation with verified information.

Therefore, we pose the following research question: How can we develop a robust methodology to quantify a model's uncertainty regarding its parametric knowledge, and further enable it to refrain from generating hallucinated responses, without internal model access or external assistance?

Current research predominantly evaluates self-consistency on the original query (Farquhar et al., 2024; Manakul et al., 2023b; Lin et al., 2024; Aichberger et al., 2024; Yadkori et al., 2024) to analyze a model's uncertainty for a single query. These approaches calculate uncertainty by sampling multiple responses to the same query and measuring consistency using entropy or other uncertainty evaluation methods across semantically clustered responses. While inconsistency about the original query in LMs often coincides with hallucination, these approaches do not necessarily capture a model's uncertainty about the veracity of its response (Zhang et al., 2023; Zhao et al., 2024; Chen et al., 2024a). A model may consistently provide an incorrect answer to the original query, while

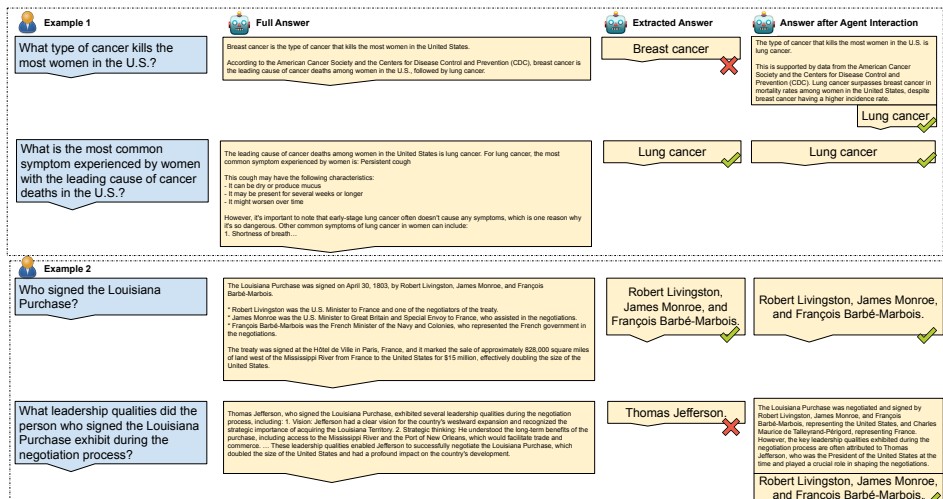

Figure 1: Two examples where an LLM shows different behaviors on diverse questions from different perspectives about the original queries.

consistently giving a correct response to varied questions from different perspectives that require the same underlying fact, or vice versa, as demonstrated in the examples in Fig. 1.

We start with a simple assumption: if a model is certain of its answer to a query, it should consistently provide the same answer across different questions that rely on the same underlying information. However, we observe that providing additional context in varied questions influences the model's behavior by exposing it to different background information, which can result in varying outcomes (Gonen et al., 2023; Sclar et al., 2024). In some cases, as shown in Example 1 of Fig. 1, the additional context helps the model better assess its own knowledge. However, in other cases, such as Example 2 in Fig. 1, it introduces confusion.

Previous studies have demonstrated that allowing LLMs to revise their responses (Kadavath et al., 2022; Shinn et al., 2023), while simultaneously presenting them with diverse related contextual information (Sun et al., 2023), can improve the accuracy of their answers. Building on these intuitions, we propose to assess the model's uncertainty regarding their parametric knowledge after multi-agent interaction with the same underlying model (Xiong et al., 2023; Du et al., 2024; Feng et al., 2024) as shown in Fig. 2. Specifically, we define an agent as the same base model, but with different background knowledge, acquired by first answering a unique varied question related to the original query. The varied questions should require the model to rely on the same underlying information as the original query while introducing diverse perspectives or variations. We then encourage multiple rounds of controlled one-on-one agent interactions, allowing the agents to collaboratively refine their answers to the original query. We detail the entire agent interaction process in §3.3 and 3.4. The multi-agent interaction process exposes the model to diverse perspectives on the same original query through different agents' questions and responses, allowing it to self-correct. As shown in Fig. 2, after the agent interaction, all participating agents agree on the same answer.

We then propose DIVERSEAGENTENTROPY, which uses the weighted entropy of the agents' final answers as a reliable measure of the model's uncertainty regarding the original query. As illustrated in Fig. 2, the model's final uncertainty for this query will be 0. This approach evaluates the consistency of the model's responses to the original query across a diverse range of related questions, rather than relying solely on the original query. Additionally, we define an abstention policy to withhold responses when uncertainty is high.

In this paper, we demonstrate that our uncertainty metric, when combined with an abstention policy, effectively assesses model reliability and identifies hallucinations. Our method surpasses existing black-box, self-consistency-based uncertainty estimation methods, achieving a superior AUROC score. By sampling across different abstention rates, our method consistently delivers a 2.5% improvement in accuracy on known questions compared to self-consistency-based approaches across various types

of QA tasks. Furthermore, our approach allows for an in-depth analysis of the model's ability to consistently retrieve accurate information. Notably, we find that, even when the model possesses the correct answer to a query, it frequently fails to provide consistent responses when queried from different perspectives. This finding highlights the need for improvements in the model's retrievability of parametric knowledge. Finally, we conduct comprehensive ablation studies to examine agent interactions, providing valuable insights for future work.

## 2 RELATED WORK

**Uncertainty Estimation of LMs.** Several recent works (Farquhar et al., 2024; Yadkori et al., 2024; Lin et al., 2024; Aichberger et al., 2024) have systematically quantified LLM uncertainty using entropy over multiple sampled outputs; however, they all focus on self-consistency to the original query, which can be misleading as shown in Figure 1. Some studies attempt to verbalize LLM uncertainty (Tian et al., 2023; Xiong et al., 2024), but Xiong et al. (2024) shows that LLMs are overconfident when verbalizing their confidence. Some works measure uncertainty from the LLM's activations (Chen et al., 2024b; CH-Wang et al., 2024) while we don't have access to model internals.

**Consistency Evaluation of LMs.** Although Wang et al. (2023) demonstrates that self-consistency with a majority vote can significantly enhance reasoning in LMs and Manakul et al. (2023a) further proposes a simple sampling-based approach that can be used to fact-check the response, Zhang et al. (2023) and Zhao et al. (2024) argue that detecting factual hallucinations requires evaluating consistency across semantically equivalent questions, not just self-consistency. Additionally, Chen et al. (2024a) further illustrates that LLMs struggle to maintain compositional consistency. Therefore, our paper adopts a broader definition of consistency to better quantify the model's output certainty.

**Agent interaction for LMs.** Recent works (Xiong et al., 2023; Du et al., 2024; Feng et al., 2024) improve factuality in LMs through multi-agent cooperation or debate, primarily using cross-model agents. In contrast, we build same-model agents. The most similar setting is Feng et al. (2024), though it doesn't allow self-correction. Our method facilitates controlled interactions for simplified analysis. Future work will explore enhancing agent interactions, e.g., with persona-based variations.

## 3 METHOD

### 3.1 BACKGROUND ON NLG UNCERTAINTY ESTIMATION

We first provide background on uncertainty estimation, focusing on entropy-based evaluation, as uncertainty is commonly measured by the entropy of predictions in the existing literature (Wellmann & Regenauer-Lieb, 2012; Abdar et al., 2021). We denote $x$ and $Y$ as the input—original query—and the output—random variable $Y$. The total uncertainty for a given model $\theta$ can be understood as the predictive entropy of the output distribution:

$$U(x) = H(Y|x) = -\int p(y|x) \log\left(p(y|x)\right) dy. \tag{1}$$

If the overall uncertainty $U$ is low, the model has high confidence in its output. Since it is impractical to sample all possible answers, directly calculating Eq.1 is not feasible. Instead, in NLG, we approximate using (Malinin & Gales, 2021; Farquhar et al., 2024; Aichberger et al., 2024):

$$U(x) = H(Y|x) \approx -\sum_{y_i \in C} p(y_i|x) \log p(y_i|x). \tag{2}$$

C represents all grouped semantically different answers obtained when a model is queried $N$ times with the same input, i.e., the original query $x$. $y_i$ is one possible semantically different answer for $x$.

### 3.2 EXISTING SELF-CONSISTENCY BASED UNCERTAINTY ESTIMATION

In this section, we explain how self-consistency on a single query can be applied to approximate the model's uncertainty, along with its limitations. Existing self-consistency-based uncertainty estimation methods in the black-box setting (Kuhn et al., 2023; Farquhar et al., 2024; Lin et al., 2024; Aichberger et al., 2024) follow a similar procedure: 1) For a given input $x$, generate $N$ response samples. 2)

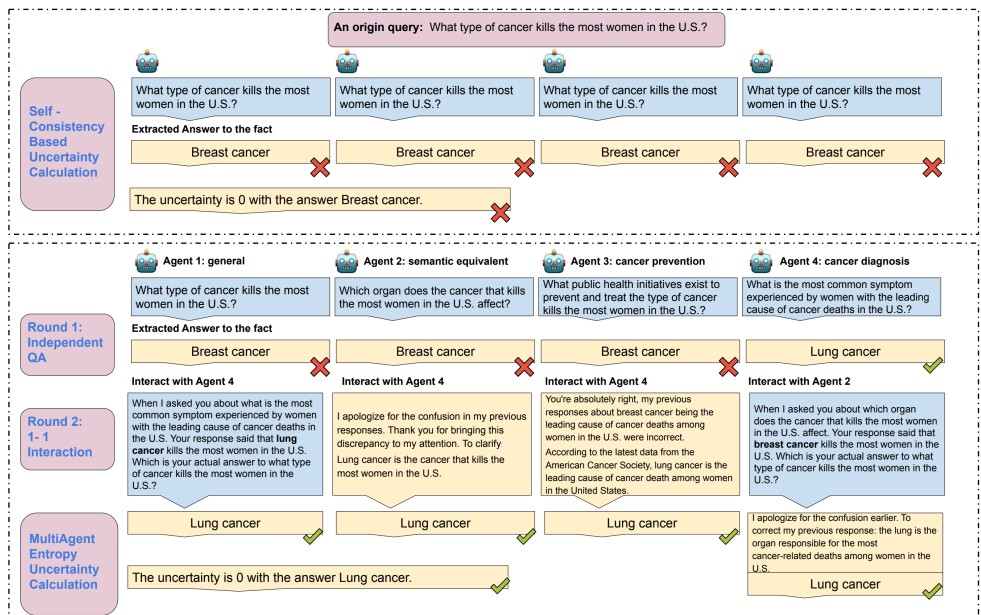

Figure 2: Our proposed DIVERSEAGENTENTROPY estimates the model's uncertainty about a single query by encouraging multi-agent interactions on varied questions related to the original query. Instead of relying on self-consistency for the original query, we analyze the model's uncertainty after these agent interactions.

Calculate pairwise similarity scores for these $N$ responses. 3) Compute an uncertainty estimate $U(x)$ using the similarity values.

Specifically, Farquhar et al. (2024) introduces semantic entropy to calculate $p(y_i|x)$ in Eq.2 as a frequency-based probability derived from repeated sampling of the original question $x$. Assume we find the semantic clusters for the sampled answers, and let each query return a possible semantically different answer $y_i \in C$. The count of times a particular $y_i$ appears as the output for the input $x$ over these $N$ queries is denoted as $c(y_i)$. Therefore, $p(y_i|x) = \frac{c(y_i)}{N}$.

Lin et al. (2024) calculates uncertainty using a weighted adjacency graph built upon semantic affinities. An affinity model $e$ maps pairs of responses to values in $[0, 1]$. Given $N$ independent samples, the model induces a symmetric adjacency matrix $W = [w_{i,j}]_{i,j=1}^N$, where $w_{i,j}$ is the mean of the pairwise similarities between response $i$ and $j$. The degree matrix is $D = [\mathbb{1}[j = i] \sum_{n=1}^N w_{n,j}]_{i,j=1}^N$, and the Laplacian $L = I - D^{-1/2}WD^{-1/2}$ has eigenvalues $\{\lambda_n\}_{n=1}^N$. The following uncertainty measures are then defined: $U_{EigV}(x) = \sum_{n=1}^N \max\{0, 1 - \lambda_n\}, U_{Degree}(x) = 1 - \frac{trace(D)}{N^2}, U_{Ecc}(x) = \|[v_1, v_2, \ldots, v_N]\|_2$ where $\{v_n\}_{n=1}^N$ are vectors associated with $L$.

As a result, regardless of the specific method used, if all $N$ responses sampled from the original query consistently output the same answer with the same semantics $y'$, the model is considered certain about the answer to the query $x$ with the lowest uncertainty. However, self-consistency alone is insufficient for accurately assessing a model's uncertainty of the original query. A model may consistently provide incorrect answers to the query but recall the correct answer when responding to varied related questions (Fig. 1, Example 1). Conversely, it may initially provide the correct answer but fail to recall it when answering related questions (Fig. 1, Example 2).

### 3.3 DIVERSEAGENTENTROPY: PROPOSED METRIC OF UNCERTAINTY

Acknowledging the limitations of commonly used self-consistency-based methods, in this section, we introduce DIVERSEAGENTENTROPY, a multi-agent interaction approach that extends beyond self-consistency to estimate the uncertainty of an LLM for a single query in a black-box setting. Our method is illustrated in Fig. 2.

Based on the two observations in Fig. 1, we first make a more robust assumption for modeling uncertainty: if a model is certain, it should consistently recall the answer to the query across a diverse collection of questions about the same query. For example, for popular queries like "What is the current capital of France?", the model is certain and will output "Paris" for any varied questions. Instead of repeatedly querying the model with the same original query $x$, we aggregate responses across a variety of diverse questions $Q = \{q_1, q_2, \ldots, q_n\}$, where the answer to the original query is required during the answering process. The set of Q will include the original query x itself, semantically equivalent questions of the original query x, and questions about different perspectives as shown in Fig. 2. The automated varied question generation process is described in detail in §3.4. We then query the model with each variant of the question $q_j$ in Q, and count the occurrence of specific semantically different answer $y_i$ to the original query. Let $c(y_i, q_j)$ denote the count of the semantically different answer $y_i$ to the original query extracted from the response to the question $q_j$. The aggregated counts across all different inputs $q_j$ are used to estimate $p(y_i|x)$ as:

$$p(y_i|x) = \frac{\sum_{j=1}^{n} c(y_i, q_j)}{\sum_{j=1}^{n} N_j}. \tag{3}$$

where $N_j$ is the number of times the model was queried with the input $q_j$, and $n$ is the total number of distinct questions related to the same original query. We set $N_j = 1$ for simpler implementation.

However, since we observe that providing additional context in varied questions influences the model's behavior, we propose an additional multi-agent interaction process to further calibrate the calculation of $p(y_i|x)$. This process allows the model to engage in self-reflection and self-correction. We create $n$ agents from the same tested model where each agent $A_j, j = 1, \ldots n$ first independently answers a unique varied question $q_j, j = 1, \ldots n$ about the original query once as its unique context background. After initial responses are generated from different agents, we will extract their answers to the original query from their responses. As demonstrated by Wang et al. (2024), a transformer-based model can perform gradient descent on common alignment objectives in an in-context manner and engage in self-correction. We then facilitate multiple rounds of collaboration between agents, specifically through one-on-one interactions, to help refine their answers to the original query, as shown in Fig. 2.

Specifically, we conduct controlled cross-play one-on-one interactions, where different agents engage with one another using fixed prompts, as illustrated in Fig. 2. The interaction is limited to a maximum of $R^*$ rounds. For each agent $A_j$ in a round, we will randomly select another agent whose answer to the original query differs for interaction. We will prioritize choosing an agent with whom the agent $A_j$ has not previously interacted. During this round of interaction, agent $A_j$ will be shown its previous conversation history, including its initial question, response, and previous interactions. Additionally, it is presented with the current round's information, which includes the other agent's unique question and its answer to the original query from the previous round. The agent $A_j$ is then prompted to decide which is the correct answer—either maintaining or changing its own response. This process mitigates the model's inconsistencies with varied questions by in-context fine-tuning, allowing the model to read diverse content from different agents' questions and responses and self-correct its answers.

Given that different agents have varying levels of response credibility, we calculate the weight $w_j$ for each agent $A_j$ in the final probability calculation. Based on ground truth independence assumption from Yadkori et al. (2024), if the model is certain about the answer to the question, the response to a prompt containing the question and previous responses to the same question is insensitive to the previous response. As a result, an agent that frequently changes its answer during these interactions is considered less reliable. Consequently, its final answer should be assigned a lower weight. We thus calculate the weight $w_j$ based on how often the agent $A_j$ changes its answer to the original query.

$$w_j = \frac{R - r_j + 1}{\sum_{j=1}^{n}(R - r_j + 1)}. \tag{4}$$

where $j = 1, \ldots, n$. We denote $R$ as the final total number of interaction rounds and $r_j$ as the number of rounds where the agent $A_j$ changes its answer during the interaction. We apply Laplace smoothing to avoid zero weight. Denote by $\mathbb{1}\{A_j = y_i\}$ whether $y_i$ is the final answer to the original query of the agent $A_j$ after the interaction. Therefore,

$$p(y_i|x) = \sum_{j=1}^{n} w_j \mathbb{1}\{A_j = y_i\}. \tag{5}$$

We can then apply Eq.2 with Eq.5 to calculate the final uncertainty as DIVERSEAGENTENTROPY. The quality of the approximation for Eq.2 is improved compared to simple self-consistency entropy in Eq.3.2, as we have better-approximated probabilities for each $p(y_i|x)$: 1) DIVERSEAGENTENTROPY enables the sampling of a broader range of potential answers by introducing varied questions with different contexts. 2) The answers retained after agent interaction are those with significant probability mass, as they represent the responses that agents consistently agree upon.

### 3.4 IMPLEMENTATION

Below we detail how to implement the above-mentioned DIVERSEAGENTENTROPY.

**Step 1: Question Generation.** Given an original query $x$, we use the same model to be tested to generate varied questions that require knowledge of the original query, ensuring that these questions are both representative and comprehensive. The question-generation process is completely automated and the detailed question-generation prompts can be found in §A.10. Specifically, we first conceptualize the original query and then sample various perspectives to ensure a comprehensive understanding. For each perspective, we generate $m$ questions that build upon the original query, tailored to that particular perspective. We filter these generated questions to ensure they strictly require knowledge of the original query to answer while avoiding the inclusion of the direct answer. We also generate $m$ semantically equivalent questions for the original query.

We select $n$ questions from the generated pool to form the final candidate set $Q$ for the agents. This set includes the original query $x$, one semantically equivalent question, and $n - 2$ questions each targeting a unique perspective. If there are insufficient unique perspectives with qualified questions, we repeat the perspective question selection process to select from existing perspectives. If not, we supplement with additional semantically equivalent questions.

**Step 2: Agent Interaction.** We follow the interaction process as mentioned in §3.3. During the interaction process, the agent $A_j$ may maintain its own answer to the single fact, accept other agent's answer or output I don't know. An answer to the original query will be extracted after each 1-1 interaction. The detailed interaction prompts are shown in §A.10. The interaction concludes under any of the following conditions: 1) unanimous agreement among all agents on the answer to the original query, 2) all agents consistently maintain their selected answer for at least two consecutive rounds, or 3) the interaction reaches the predefined maximum of $R^*$ rounds.

**Step 3: Uncertainty Score Calculation.** We follow Eq.5 to calculate the probability for each semantically different answer of the agents. We can then calculate the final uncertainty as in Eq.2. While acknowledging that our method is more resource-intensive than self-consistency-based approaches, we provide a detailed cost analysis in §A.1.

### 3.5 SCORE-BASED ABSTENTION POLICY

The uncertainty derived above can be used as a score to assess whether the model's answer to a given query can be trusted and to detect potential hallucinations. We then introduce an abstention policy with a threshold parameter. This policy triggers abstention when the uncertainty score exceeds the threshold (see §4.1 for proposed method variants). If the policy does not abstain, the answer with the highest calculated probability is provided.

## 4 EXPERIMENT

### 4.1 EXPERIMENT SETTING

**Evaluation Models.** We evaluate on Llama-3-70b-Instruct (AI@Meta, 2024) and Claude-3-Sonnet (Anthropic, 2024).

**Datasets.** We consider five different datasets under three categories. See §A.2 for a detailed description of the datasets. **Entity-centric QA**: we randomly sample from PopQA (Mallen et al., 2023) for 1) PopQA popular with popular entities and 2) PopQA less popular with less popular entities. **General QA**: 3) TruthfulQA (Lin et al., 2022). We only sample questions about clear facts instead of opinions. 4) FreshQA (Vu et al., 2023). We adopt the 07112024 version and further filter

always-changing questions. **False assumption QA**: 5) FalseQA (Hu et al., 2023). All questions in the dataset contain false assumptions and we remove all the WHY questions.

**Metrics.** Following prior work (Lin et al., 2022; Farquhar et al., 2024), we assess uncertainty score by treating uncertainty estimation as whether to trust an answer to a question. We first evaluate the AUROC score for the entropy-based methods. Our main experiments focus on evaluating the accuracy of DIVERSEAGENTENTROPY for hallucination detection. We evaluate the model's performance after applying the abstention policy based on the uncertainty score: 1) *accuracy*, the percentage of correct responses, i.e, the answer from the model matches the gold answer, among the questions where the model does not abstain; 2) *abstention rate*, the percentage of questions where the method abstains; 3) *correctness score*, the percentage of correct responses among all the questions; 4) *truthfulness score* (Lin et al., 2022), the percentage of correct or abstained responses among all the questions. We further analyze the accuracy-recall (AR) trade-off across various methods and datasets. Here, recall is the percentage of questions where the method does not abstain, i.e., recall = 1 - abstention rate.

**Baselines.** We adopt four black-box uncertainty estimation baselines as described in §3.2 to evaluate the calibration of DIVERSEAGENTENTROPY and the model is prompted to answer the original question 5 times: 1) Self-consistency with SemanticEntropy (SC SE) (Farquhar et al., 2024). We describe the detailed implementations in §A.3. Three baselines with affinity graph (Lin et al., 2024): 2) Self-consistency with Eccentricity (SC Ecc). 3) Self-consistency with the Degree Matrix (SC Degree). 4) Self-consistency with Eigenvalues (SC EigV).

We adopt seven baselines for hallucination detection. **Greedy-based baselines**: 1) Greedy: the model is prompted to answer the original query once with greedy decoding. 2) Self-Evaluation (Kadavath et al., 2022) The model first outputs a greedy answer and then is asked to reevaluate its own answer. 3) Self-evaluation w many samples (Kadavath et al., 2022). 5 answers including the greedy answer are generated in total, and then the model is asked about the validity of the greedy sample. 4) Multiple-Recite (Sun et al., 2023). The model is prompted to generate multiple related paragraphs from its parametric knowledge before answering the question. **Sampling-based baselines:** 5) Self-consistency (SC) (Wang et al., 2023): the model answers the query 5 times, and we accept the majority answer or abstain if no answer appears at least 3 times. 6) Consistency with semantically equivalent questions (SeQ) (Zhang et al., 2023; Zhao et al., 2024): the model is prompted to answer 5 semantically equivalent questions about the same original query. 7) Consistency with diverse questions (DiverseQ): the model is prompted to answer 5 diverse questions about the same original query generated the same as in §3.4. Note that we evaluate the semantic equivalence of answers to cluster responses for all sampling-based baselines.

**Proposed method variants.** We adopt two variants of DIVERSEAGENTENTROPY where we have 5 agents, i.e., 5 varied questions: 1) Agent (Loose Majority Vote): We abstain when the uncertainty score exceeds the threshold, calculated as the entropy of 3 answers with probabilities of 0.6, 0.2, and 0.2. 2) Agent: We use a stricter majority vote, abstaining when the uncertainty score exceeds the threshold, calculated as the entropy of 2 answers with probabilities of 0.6 and 0.4. We further explain the intuitions behind the choices in §A.4.

### 4.2 EVALUATION OF DIVERSEAGENTENTROPY AND ITS USAGE

In this section, we aim to assess whether our proposed method reliably indicates the model's ability to provide more accurate responses or appropriately refuse to answer when necessary. We also evaluate the model's effectiveness in retrieving correct knowledge consistently.

**DIVERSEAGENTENTROPY is more calibrated than self-consistency-based uncertainty estimation.** We present the AUROC score for comparison between self-consistency-based uncertainty estimation methods and our DIVERSEAGENTENTROPY in Table 1. The results indicate that our proposed method is better calibrated, as evidenced by the highest AUROC score. We further detail the calibration of the proposed uncertainty score in Appendix Fig. 6 where the uncertainty scores are grouped into ten equally sized bins and we calculate the correctness of predictions in each bin. For all models, correctness is inversely correlated with the uncertainty score and our method demonstrates better calibration compared to SemanticEntropy.

**DIVERSEAGENTENTROPY-based abstention policy effectively detects hallucinations.** We show in Table 2 that the uncertainty estimated by the proposed DIVERSEAGENTENTROPY has a better

| Model | FalseQA | FreshQA | TruthfulQA | PopQA_less_popular | PopQA_popular | All |
|---|---|---|---|---|---|---|
| | | | *Claude-3-Sonnet* | | | |
| SC (Ecc) | 0.711 | 0.702 | 0.548 | 0.821 | 0.671 | 0.766 |
| SC (Degree) | 0.713 | 0.704 | 0.550 | 0.855 | 0.674 | 0.771 |
| SC (EigV) | 0.713 | 0.703 | 0.550 | 0.851 | 0.673 | 0.771 |
| SC (SE) | 0.753 | 0.694 | 0.568 | 0.887 | 0.693 | 0.792 |
| Agent | **0.802** | **0.836** | **0.624** | **0.947** | **0.725** | **0.833** |
| | | | *Llama-3-70b-Instruct* | | | |
| SC (Ecc) | 0.628 | 0.660 | 0.488 | 0.716 | 0.594 | 0.644 |
| SC (Degree) | 0.629 | 0.662 | 0.486 | 0.704 | 0.595 | 0.645 |
| SC (EigV) | 0.629 | 0.664 | 0.486 | 0.707 | 0.595 | 0.645 |
| SC (SE) | 0.673 | 0.632 | 0.545 | 0.737 | 0.624 | 0.694 |
| Agent | **0.673** | **0.697** | **0.592** | **0.753** | **0.651** | **0.713** |

Table 1: Comparison of AUROC scores between self-consistency based methods and our DIVERSEAGENTENTROPY (Agent) across different QA datasets. Our method is more calibrated.

diagnostic ability to identify whether the model is hallucinating. It is more effective in abstaining from answering when the model is uncertain and thus more accurate in outputting correct answers when the model does not abstain. Also, our agent method has the highest correctness score and truthfulness score, further indicating its advantages over other baselines. We present the performance of individual datasets in §A.6. Fig. 3 presents Accuracy-Recall (AR)-curves for the baselines and DIVERSEAGENTENTROPY across all data. Detailed performance for each dataset is provided in Appendix Fig. 8. The results clearly demonstrate that our proposed method outperforms all baselines. Among the recall rates where all methods can be applied, our proposed method has the highest accuracy.

| Method | Claude-3-Sonnet | | | | Llama-3-70b-Instruct | | | |
|---|---|---|---|---|---|---|---|---|
| | Acc | Ab-R | Correct | TruthF | Acc | Ab-R | Correct | TruthF |
| Greedy | 0.808 | 0.126 | 0.707 | 0.832 | 0.775 | 0.008 | 0.769 | 0.777 |
| Self-Reflect | 0.826 | 0.131 | 0.718 | 0.849 | 0.783 | 0.030 | 0.760 | 0.790 |
| Self-Eval w Samples | 0.814 | 0.141 | 0.700 | 0.840 | 0.754 | 0.020 | 0.739 | 0.759 |
| Multiple-Recite | 0.779 | 0.114 | 0.690 | 0.804 | 0.715 | 0.010 | 0.708 | 0.717 |
| SC (3/5) | 0.823 | 0.129 | 0.717 | 0.846 | 0.794 | 0.035 | 0.766 | 0.801 |
| SeQ | 0.815 | 0.149 | 0.693 | 0.842 | 0.818 | 0.084 | 0.749 | 0.833 |
| DiverseQ | 0.858 | 0.342 | 0.564 | 0.906 | 0.811 | 0.121 | 0.713 | 0.834 |
| Agent (Loose Majority Vote) | 0.852 | 0.142 | **0.731** | 0.873 | 0.826 | 0.055 | **0.780** | 0.835 |
| Agent | **0.883** | 0.216 | 0.692 | **0.908** | **0.841** | 0.084 | 0.770 | **0.854** |

Table 2: Performance evaluation of different models on all data points. Acc refers to accuracy. Ab-R refers to abstention rate. Correct refers to correctness score. TruthF refers to truthfulness score.

**The retrievability of parametric knowledge remains unsatisfying.** We demonstrate that even when the model knows the correct answer based on our proposed uncertainty evaluation, they initially fail to consistently retrieve the same response across different contexts or scenarios, i.e. when answering varied questions. We conduct both quantitative and qualitative analyses to assess whether the model effectively retrieves accurate knowledge with the assistance of our proposed method. We particularly focus on instances where all agents agree on the same gold answer after interaction, as this consensus indicates that the model has correctly identified the answer to the query.

We begin with a quantitative analysis to evaluate the model's initial performance by calculating the average percentage of incorrect answers to the original query in the first round. This metric reflects how often the model fails to retrieve the correct answer initially, before any interaction. The results in Fig. 3 confirm that models are not always reliable in providing consistent answers to the same question across different contexts. This issue is particularly pronounced when the original queries are less popular as PopQA less popular, or more general, as observed in FreshQA and TruthfulQA. We further conduct a qualitative analysis by sampling 45 instances from the same pool, focusing on cases where the agents do not agree on the gold answer in the first round. The authors manually annotate the reasons for the model's failure to retrieve the correct answer without interaction. We observe that the model is more likely to generate a different response, even when it knows the correct answer, under

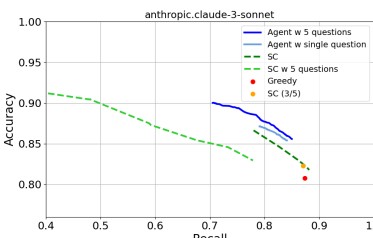 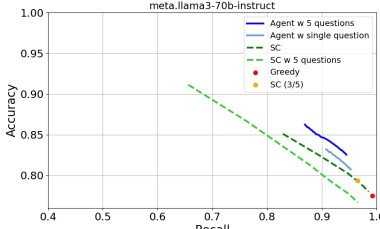

Figure 3: AR-curves for the tested methods across all data. Agent w 5 questions refers to our DIVERSEAGENTENTROPY. SC refers to SC (SE). SC w 5 questions refers to calculating entropy using the agents' diverse questions without agent interaction.

several conditions: 1) 42% of cases occur when the added context in the varied question significantly diverges from the original query, 2) 22% when an incorrect answer is more popular within the context of the original query, and 3) 20% when the additional context is more closely related to a different possible answer to the original query. Examples of each scenario are provided in §A.7.

These findings highlight the need for systematic research into how often models rely on semantic associations from pre-training data, overlooking other crucial content in the question (Zhang et al., 2024; Li et al., 2024). The behavior we have observed in the models can significantly undermine the credibility of their outputs. Potential solutions include fine-tuning/ knowledge editing models with varied questions related to the same query simultaneously.

## 4.3 ANALYSIS OF THE PROPOSED DIVERSEAGENTENTROPY

**Both diverse question generation and agent interaction are key components for performance boost.** In Fig. 3, Comparing our proposed method without interaction (SC with 5 questions) and the proposed method highlights the effectiveness of agent interaction. Furthermore, comparing the use of the original query alone for agent interaction (agent with a single question) to our proposed method demonstrates the effectiveness of diverse question generation. We also present the ratios of initially correct responses that become incorrect after agent interaction (Wrong), and initially incorrect responses that become correct after interaction (Correct) for each dataset in Table 4. This analysis further demonstrates the effectiveness of agent interaction. The results show that our DIVERSEAGENTENTROPY method enables the model to correct a significant number of initially incorrect responses while rarely causing initially correct answers to become incorrect.

| Dataset | Claude-3-Sonnet | Llama-3-70b-Instruct |
|---|---|---|
| PopQA pop | 0.114 | 0.118 |
| PopQA less pop | 0.193 | 0.207 |
| FalseQA | 0.154 | 0.154 |
| TruthfulQA | 0.296 | 0.330 |
| FreshQA | 0.167 | 0.175 |

Table 3: Average percentage of incorrect answers to the query in the first round without agent interaction, in cases where all agents agree on the correct answer after agent interaction.

| Dataset | Claude-3-Sonnet | | Llama-3-70b-Instruct | |
|---|---|---|---|---|
| | Wrong | Correct | Wrong | Correct |
| PopQA pop | 0.152 | 0.487 | 0.200 | 0.545 |
| PopQA less pop | 0.061 | 0.179 | 0.055 | 0.300 |
| FalseQA | 0.000 | 0.042 | 0.088 | 0.140 |
| TruthfulQA | 0.035 | 0.568 | 0.150 | 0.605 |
| FreshQA | 0.089 | 0.302 | 0.086 | 0.381 |

Table 4: The ratios of instances where initially correct responses become incorrect, and initially incorrect responses become correct.

**The number of agents.** We analyze the impact of agents number. In Fig. 4 and Appendix Fig. 9, we increase the number of agents, limiting interactions to 4 rounds. Performance improves with more agents but shows minimal gains beyond 4 agents, suggesting 5 agents are sufficient.

**The rounds of interactions.** We analyze the impact of the number of interaction rounds in Fig. 4 and Appendix Fig. 10, with the number of agents fixed at 5, increasing the rounds of interaction generally leads to improved performance.

**Format of agent interaction.** We examine whether agents should engage in one-on-one interactions or group interactions, where in group settings, each agent can view the unique questions and answers of all other agents. Our findings, presented in Fig. 5 and Appendix Fig. 11, The results indicate that

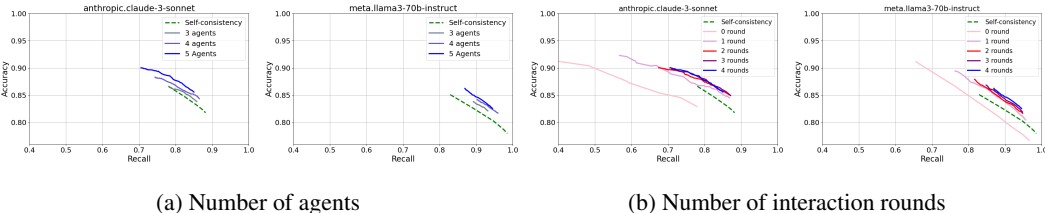

(a) Number of agents         (b) Number of interaction rounds

Figure 4: The figures show that more agents and more rounds of interaction improve performance.

one-on-one interactions outperform group interactions. In our analysis of 30 incorrect examples from each model, we identified two primary error types: (1) 50% of errors occurred when agents were influenced by the majority's incorrect answer, and (2) 15% occurred when agents concluded that the question had no valid answer or was based on a false premise due to conflicting responses. This analysis further demonstrates that agents are more easily influenced by dominant incorrect information, reinforcing the importance of using one-on-one interactions for single-query uncertainty checking, as it allows the model to be exposed to diverse information while maintaining its ability to apply independent reasoning.

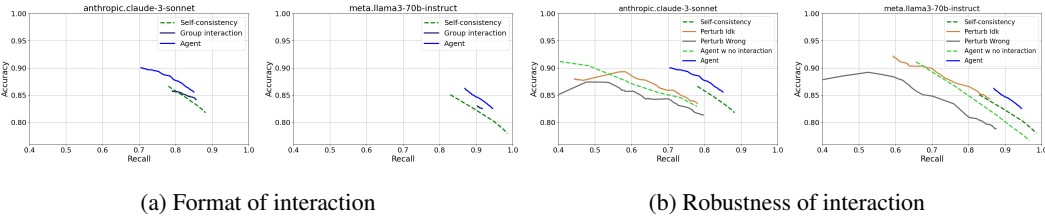

(a) Format of interaction         (b) Robustness of interaction

Figure 5: We present figures that analyze agent behaviors during interactions.

**Robustness of agent interaction.** Finally, we analyze the susceptibility of agents to being misled when one agent consistently provides the most plausible incorrect answer or repeatedly responds with "I don't know.". According to the results shown in Fig. 5 and Appendix Fig. 12, the overall performance of the agents deteriorates in both scenarios, indicating that the model is influenced by consistent misleading information, while the agents suffer more from the wrong answer setting. We also observe that agents being influenced can be monitored by tracking how often they continuously flip between two answers, suggesting future work on refining uncertainty estimation metrics in multi-agent interactions through a more detailed analysis of agent behavior.

**Limitations of DIVERSEAGENTENTROPY.** Exploring beyond simple QA sheds light on the limitations of our proposed method. We detail the analysis in §A.9. Unlike simple questions, evaluating varied questions is more effective for complex ones, further demonstrating that uncertainty is best analyzed through consistency across varied questions rather than self-consistency on a single query. We observe that agent interaction can sometimes confuse the model, as agents often prematurely suggest a question is invalid. This motivates future research to develop more advanced interaction formats for handling complex questions. A possible solution is to include a summarizer or meta-judge (Chan et al., 2023) to track agents' overall understanding of the query.

## 5 CONCLUSION

Accurately determining the uncertainty of LLMs in response to a single query in a black-box setting is challenging. In this paper, we propose a novel method, DIVERSEAGENTENTROPY, for quantifying an LLM's uncertainty based on the consistency of responses across diverse questions after multi-agent interaction. Our method overcomes the limitations of self-consistency-based uncertainty estimation and delivers superior performance in detecting hallucinations. Additionally, we show that the model's ability to retrieve parametric knowledge still requires improvement.

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

# A   Appendix

## A.1   Cost Analysis

We present a detailed cost analysis on our proposed DiverseAgentEntropy. In a self-consistency-based method, we typically sample a simple query 5 times, requiring 5 API calls. For our method, starting with a single query, we need 1 API call for question conceptualization, 1 for perspective generation, and 5 for generating questions under different perspectives. During agent interaction (with 5 agents), each agent answers their assigned question, followed by multiple interaction rounds. Assuming an average of 2.5 rounds, agent interaction requires 20 calls. In total, our method averages 25 API calls, making it 5 times more expensive than the self-consistency method. However, we would like to emphasize that in high-stakes applications where correctness is prioritized over cost, our calibrated uncertainty score can provide users with a reliable measure of how much they can trust the model's output. Additionally, the chosen answers after applying the abstention policy are more accurate. The varied questions generated by our method can also be utilized for fine-tuning or training purposes. The intermediate results generated by our method, including varied questions and the self-reflection interaction processes, can be further leveraged to create synthetic data for finetuning or training LLMs. Future work can explore ways to maintain the same level of performance while reducing costs. This could involve using fewer but higher-quality questions from diverse perspectives and minimizing the number of interaction rounds.

## A.2   Datasets statistics

| Dataset Type | Dataset Name | #Data | Example |
|---|---|---|---|
| Entity-centric QA | PopQA_less_popular | 459 | What is Geeling Ng's occupation? |
| | PopQA_popular | 452 | What is the capital of Hungary? |
| General QA | TruthfulQA | 219 | What type of cancer kills the most women in the U.S.? |
| | FreshQA | 283 | What's the city flower of Shanghai? |
| False assumption QA | FalseQA | 1867 | Which planet is larger, Mars or Moon? |

Table 5: Detailed statistics of the datasets used in the paper.

We consider five different datasets under three categories. We present the detailed dataset statistics in Appendix Table 5. **Entity-centric QA**: we randomly sample data from PopQA (Mallen et al., 2023) where each question is created by converting a knowledge tuple retrieved from Wikidata using a template. 1) PopQA popular. We sample questions where the entity's popularity is larger than $10^4$ as the criteria in the original paper. 2) PopQA less popular. We sample questions where the entity's popularity is lower than $10^4$. **General QA**: 3) TruthfulQA (Lin et al., 2022). Note that not all data in TruthfulQA tests about factual questions. We choose examples only from categories: Law, Sociology, Health, History, and Language, focusing on clear facts instead of opinions. 4) FreshQA (Vu et al., 2023). We adopt the 07112024 version and select one-hop slow-changing or never-changing data points where the effective year is before 2022 to avoid the temporal influence. **False assumption QA**: 5) FalseQA (Hu et al., 2023). All questions in the dataset contain false assumptions and we remove all the WHY questions.

## A.3   Implementation of the baselines

Note that we assess the semantic equivalence of answers to cluster responses for all sampling-based baselines and our proposed method variants. Therefore, SC(SE) is SemanticEntropy (Kuhn et al., 2023; Farquhar et al., 2024). However, instead of using the bidirectional entailment clustering algorithm proposed in semantic entropy, we directly cluster all sampled answers into semantically equivalent sets with Llama-3-70b-Instruct. We manually checked the accuracy of this LLM-based clustering on 300 instances and found the accuracy to be 98%, which is higher than the human sanity check accuracy reported in their original paper. We further present the cost, i.e., the number of inference calls for all the baselines in Appendix Table 6.

| Model | Cost |
|---|---|
| uncertainty estimation methods | |
| SC (Ecc) | 5 |
| SC (Degree) | 5 |
| SC (EigV) | 5 |
| SC (SE) | 6 |
| hallucination detection/ direct inferernce methods | |
| Greedy | 1 |
| Self-Reflect | 2 |
| Self-Eval w Samples | 6 |
| Multiple-Recite | 2 |
| SC (3/5) | 6 |
| SeQ | 7 |
| diverseQ | 13 |
| Agent | 25 |

Table 6: Comparison of cost across all the methods. Specifically, we present the number of API calls.

## A.4 THRESHOLDS FOR THE ABSTENTION POLICY

We adopt two variants of DIVERSEAGENTENTROPY where we have 5 agents, i.e., 5 varied questions: 1) Agent (Loose Majority Vote): We abstain when the uncertainty score exceeds the threshold, calculated as the entropy of 3 answers with probabilities of 0.6 (3/5), 0.2 (1/5), and 0.2 (1/5). This setup implies that at least one answer still has a majority (60%, 3/5 chance). 2) Agent: We use a stricter majority vote, abstaining when the uncertainty score exceeds the threshold, calculated as the entropy of 2 answers with probabilities of 0.6 (3/5) and 0.4 (2/5). This is the strictest majority vote threshold. The two variants balance flexibility and conservatism in decision-making: the loose majority vote allows for more uncertainty with three answer probabilities, making it suitable for situations with acceptable disagreement but where one answer is still dominant. In contrast, the stricter majority vote, using two-answer probabilities, ensures abstention in cases where only minor uncertainty can be tolerated.

## A.5 PERFORMANCE EVALUATION FOR CALIBRATION

We show the calibration of the proposed uncertainty score in Appendix Fig. 6. For all models, correctness is inversely correlated with the uncertainty score We can see from the figure that our method is more well calibrated than the best self-consistency-based uncertainty score, i.e., SemanticEntropy. Additionally, Claude-3-Sonnet achieves higher correctness above the majority vote threshold compared to Llama-3-70b-Instruct. This suggests that a larger threshold can be set for more capable models, enabling a lower abstention rate while maintaining similarly high correctness.

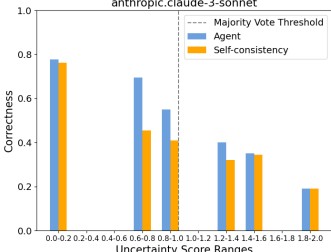 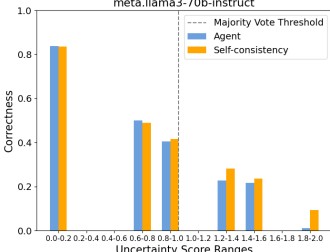

Figure 6: Calibration of the uncertainty scores. The uncertainty scores are grouped into ten equally sized bins and we calculate the correctness of predictions in each bin.

## A.6 PERFORMANCE EVALUATION FOR HALLUCINATION DETECTION ON INDIVIDUAL DATASETS.

We present the individual dataset performance on the two models in Appendix Table 7 and Appendix Table 8 respectively. We present Accuracy-Recall (AR)-curves for both the baselines and the proposed method on individual datasets in Appendix Fig. 8.

| Method | TruthfulQA | | | | FreshQA | | | | FalseQA | | | | PopQA popular | | | | PopQA less popular | | | |
|---|---|---|---|---|---|---|---|---|---|---|---|---|---|---|---|---|---|---|---|---|
| | Acc | Ab-R | TruthF | Correct | Acc | Ab-R | TruthF | Correct | Acc | Ab-R | TruthF | Correct | Acc | Ab-R | TruthF | Correct | Acc | Ab-R | TruthF | Correct |
| Greedy | 0.723 | 0.059 | 0.680 | 0.739 | 0.777 | 0.064 | 0.727 | 0.791 | 0.891 | 0.093 | 0.809 | 0.901 | 0.824 | 0.037 | 0.793 | 0.830 | 0.344 | 0.420 | 0.199 | 0.619 |
| Self-Reflect | 0.731 | 0.082 | 0.671 | 0.753 | 0.770 | 0.032 | 0.746 | 0.777 | 0.888 | 0.066 | 0.829 | 0.895 | 0.839 | 0.098 | 0.768 | 0.866 | 0.470 | 0.520 | 0.226 | 0.746 |
| Self-Eval w Samples | 0.725 | 0.087 | 0.662 | 0.749 | 0.709 | 0.064 | 0.728 | 0.664 | 0.879 | 0.077 | 0.812 | 0.889 | 0.812 | 0.059 | 0.773 | 0.832 | 0.482 | 0.562 | 0.212 | 0.774 |
| Recitation | 0.724 | 0.073 | 0.671 | 0.744 | 0.743 | 0.049 | 0.707 | 0.707 | 0.839 | 0.071 | 0.780 | 0.851 | 0.828 | 0.039 | 0.795 | 0.834 | 0.366 | 0.431 | 0.208 | 0.639 |
| SC (3/5) | 0.682 | 0.037 | 0.658 | 0.694 | 0.777 | 0.028 | 0.755 | 0.783 | 0.887 | 0.063 | 0.831 | 0.894 | 0.833 | 0.059 | 0.784 | 0.843 | 0.440 | 0.577 | 0.186 | 0.763 |
| SeQ | 0.782 | 0.183 | 0.639 | 0.822 | 0.814 | 0.163 | 0.681 | 0.844 | 0.888 | 0.099 | 0.820 | 0.899 | 0.852 | 0.061 | 0.800 | 0.861 | 0.309 | 0.420 | 0.186 | 0.606 |
| diveseQ | 0.739 | 0.261 | 0.545 | 0.807 | 0.856 | 0.216 | 0.671 | 0.887 | 0.874 | 0.302 | 0.610 | 0.912 | 0.891 | 0.193 | 0.730 | 0.923 | 0.714 | 0.777 | 0.159 | 0.936 |
| Agent (Loose Majority Vote) | 0.740 | 0.078 | 0.683 | 0.761 | 0.826 | 0.085 | 0.756 | 0.841 | 0.907 | 0.080 | 0.834 | 0.914 | 0.852 | 0.059 | 0.814 | 0.873 | 0.537 | 0.546 | 0.243 | 0.790 |
| Agent | 0.753 | 0.128 | 0.656 | 0.784 | 0.879 | 0.184 | 0.717 | 0.901 | 0.924 | 0.139 | 0.795 | 0.935 | 0.883 | 0.144 | 0.768 | 0.911 | 0.611 | 0.670 | 0.201 | 0.872 |

Table 7: Performance comparison on various datasets for Claude-3-Sonnet. Acc refers to accuracy, Ab-R refers to abstention rate, TruthF refers to truthfulness, Correct refers to correctness.

| Method | TruthfulQA | | | | FreshQA | | | | FalseQA | | | | PopQA popular | | | | PopQA less popular | | | |
|---|---|---|---|---|---|---|---|---|---|---|---|---|---|---|---|---|---|---|---|---|
| | Acc | Ab-R | TruthF | Correct | Acc | Ab-R | TruthF | Correct | Acc | Ab-R | TruthF | Correct | Acc | Ab-R | TruthF | Correct | Acc | Ab-R | TruthF | Correct |
| Greedy | 0.709 | 0.027 | 0.690 | 0.717 | 0.784 | 0.000 | 0.784 | 0.784 | 0.858 | 0.003 | 0.855 | 0.859 | 0.856 | 0.002 | 0.854 | 0.856 | 0.367 | 0.029 | 0.356 | 0.385 |
| Self-Reflect | 0.702 | 0.018 | 0.689 | 0.708 | 0.748 | 0.018 | 0.735 | 0.753 | 0.871 | 0.011 | 0.861 | 0.872 | 0.826 | 0.009 | 0.832 | 0.841 | 0.386 | 0.146 | 0.330 | 0.476 |
| Self-Eval w Samples | 0.670 | 0.046 | 0.639 | 0.685 | 0.721 | 0.000 | 0.721 | 0.721 | 0.853 | 0.022 | 0.834 | 0.856 | 0.819 | 0.002 | 0.817 | 0.819 | 0.336 | 0.033 | 0.325 | 0.358 |
| Recitation | 0.707 | 0.018 | 0.694 | 0.712 | 0.705 | 0.018 | 0.693 | 0.710 | 0.785 | 0.009 | 0.778 | 0.787 | 0.782 | 0.002 | 0.780 | 0.782 | 0.363 | 0.013 | 0.358 | 0.372 |
| SC (3/5) | 0.619 | 0.018 | 0.607 | 0.626 | 0.791 | 0.018 | 0.777 | 0.795 | 0.880 | 0.012 | 0.869 | 0.881 | 0.848 | 0.013 | 0.837 | 0.850 | 0.408 | 0.170 | 0.338 | 0.509 |
| SeQ | 0.681 | 0.116 | 0.602 | 0.718 | 0.769 | 0.066 | 0.718 | 0.784 | 0.915 | 0.064 | 0.857 | 0.921 | 0.828 | 0.034 | 0.800 | 0.834 | 0.437 | 0.215 | 0.343 | 0.558 |
| diverseQ | 0.676 | 0.155 | 0.571 | 0.763 | 0.798 | 0.088 | 0.728 | 0.813 | 0.865 | 0.071 | 0.803 | 0.874 | 0.869 | 0.065 | 0.825 | 0.891 | 0.489 | 0.389 | 0.299 | 0.688 |
| Agent (Loose Majority Vote) | 0.750 | 0.050 | 0.712 | 0.763 | 0.806 | 0.035 | 0.777 | 0.813 | 0.894 | 0.026 | 0.870 | 0.897 | 0.868 | 0.011 | 0.872 | 0.883 | 0.471 | 0.235 | 0.361 | 0.595 |
| Agent | 0.752 | 0.078 | 0.694 | 0.772 | 0.831 | 0.078 | 0.767 | 0.845 | 0.899 | 0.037 | 0.865 | 0.903 | 0.875 | 0.026 | 0.865 | 0.891 | 0.508 | 0.343 | 0.334 | 0.677 |

Table 8: Performance comparison on Llama-3-70b-Instruct for multiple datasets. Acc refers to accuracy, Ab-R refers to abstention rate, TruthF refers to truthfulness, Correct refers to correctness.

## A.7 ERROR ANALYSIS FOR THE RETRIEVABILITY OF PARAMETRIC KNOWLEDGE FOR THE MODELS.

We conduct the error analysis for the retrievability of parametric knowledge for the models with 45 examples, 23 sampled from Llama-3-70b-Instruct and 22 sampled from Claude-3-Sonnet. We observe similar behaviors in both models. As we discuss we observe that the model is more likely to generate a different response, even when it knows the correct answer, under these three conditions: 1) Example 1 sampled from Llama-3-70b-Instruct in Appendix Table 9: the added context in the varied question significantly diverges from the original query. The chosen varied question is the least similar question to the original query among the 5 varied questions according to the score of SentenceBert. 2) In Example 2, sampled from Llama-3-70b-Instruct in Appendix Table 9, an incorrect answer is more popular within the context of the original query. For instance, Cristiano Ronaldo is a more well-known football player compared to Ali Daei. 3) Example 3, sampled from Claude-3-Sonnet in Appendix Table 9, illustrates a case where the additional context is more closely related to a different possible answer. In this instance, the model is distracted by the "Yangtze River Delta region" mentioned in the varied question. Note that after agent interaction, the models answer all the questions correctly.

## A.8 PERFORMANCE OF ABLATION STUDIES

We present the performance of two models across all datasets for different ablations studies. The results can be referred to from Appendix Fig. 4 to Appendix Fig. 12.

## A.9 DISCUSSION OF EXTENSION TO COMPLEX QUESTIONS WITH SHORT-FORM ANSWER

Exploring beyond simple QA sheds light on the limitations of our proposed method. We analyze our proposed method on 450 randomly sampled instances from HotpotQA (Yang et al., 2018) in Fig. 7 where all the data are multi-hop questions. Opposite to the behaviors on simple questions, evaluating directly on varied questions is very effective for complex questions whereas agent interaction may confuse the model. Our error analysis identifies two predominant types of errors: 1) 40% of the errors occur when agents converge on the belief that the question involves a false assumption, lacks

| Example | Original Query / Gold Answer | Full Answer (Varied Question) / Extracted Answer |
|---|---|---|
| **Example 1** | **Original Query:** What film starred Brad Pitt's first leading role? | **Gold Answer:** The Dark Side of the Sun |
| | **Varied Question:** What film starred Brad Pitt's first leading role? | **Full Answer:** Thelma & Louise 
 **Extracted Answer:** Thelma & Louise |
| | **Varied Question:** What was the influence of the film starring Brad Pitt's first leading role on the development of the Gothic horror genre in cinema? | **Full Answer:** You're likely referring to the 1989 film "Dark Side of the Sun", which was not a Gothic horror film. Brad Pitt's first leading role was actually in the 1988 film "Dark Side of the Sun". 
 **Extracted Answer:** The Dark Side of the Sun |
| **Example 2** | **Original Query:** Who was the first football player to score one hundred international goals? | **Gold Answer:** Ali Daei |
| | **Varied Question:** Who was the first football player to score one hundred international goals? | **Full Answer:** The first football player to score one hundred international goals is Ali Daei of Iran. 
 **Extracted Answer:** Ali Daei |
| | **Varied Question:** What kind of endorsement opportunities and commercial deals were offered to the first football player to score one hundred international goals? | **Full Answer:** You're referring to Cristiano Ronaldo, who became the first male football player to score 100 international goals in 2021. After achieving this milestone, he received numerous endorsement opportunities and commercial deals. Some of these include: 1. Nike: A long-term contract extension with a significant increase in his annual salary. 2. Tag Heuer:... 
 **Extracted Answer:** Cristiano Ronaldo |
| **Example 3** | **Original Query:** What is the city flower of Shanghai? | **Gold Answer:** Magnolia denudata |
| | **Varied Question:** What is the city flower of Shanghai? | **Full Answer:** The city flower of Shanghai is the white peony. Known for its beauty, elegance, and cultural significance in China, the white peony symbolizes prosperity, grace, and peace. 
 **Extracted Answer:** White Peony |
| | **Varied Question:** How does the city flower of Shanghai reflect the city's historical and cultural connection to the Yangtze River Delta region? | **Full Answer:** The city flower of Shanghai is the White Peony (Paeonia lactiflora), which reflects the city's historical and cultural connection to the Yangtze River Delta region in several ways: 1. Native origin: The White Peony is native to the Yangtze River Delta region, where Shanghai is located. This emphasizes the city's deep roots in the region's natural environment. .. 
 **Extracted Answer:** White Peony (Paeonia lactiflora) |

Table 9: Error analysis with 3 examples for the retrievability of parametric knowledge for models. Note that after agent interaction, the models answer all the questions correctly.

an answer, or contains unspecified entities, and 2) 10% of errors arise when agents hesitate between two answers, one of which is the correct answer. The results indicate that when the initial query is complex, the agents are more inclined to take a shortcut by suggesting there is an issue with the question itself, as a means to avoid inconsistencies in its answers. A possible solution is to include a summarizer or meta-judge (Chan et al., 2023) to track agents' overall understanding of the query.

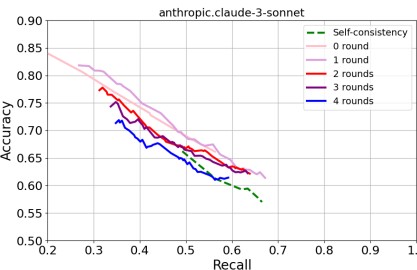 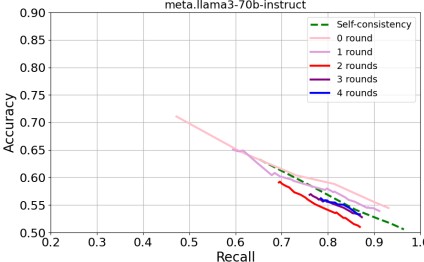

Figure 7: Performance of our Agent method on HotpotQA.

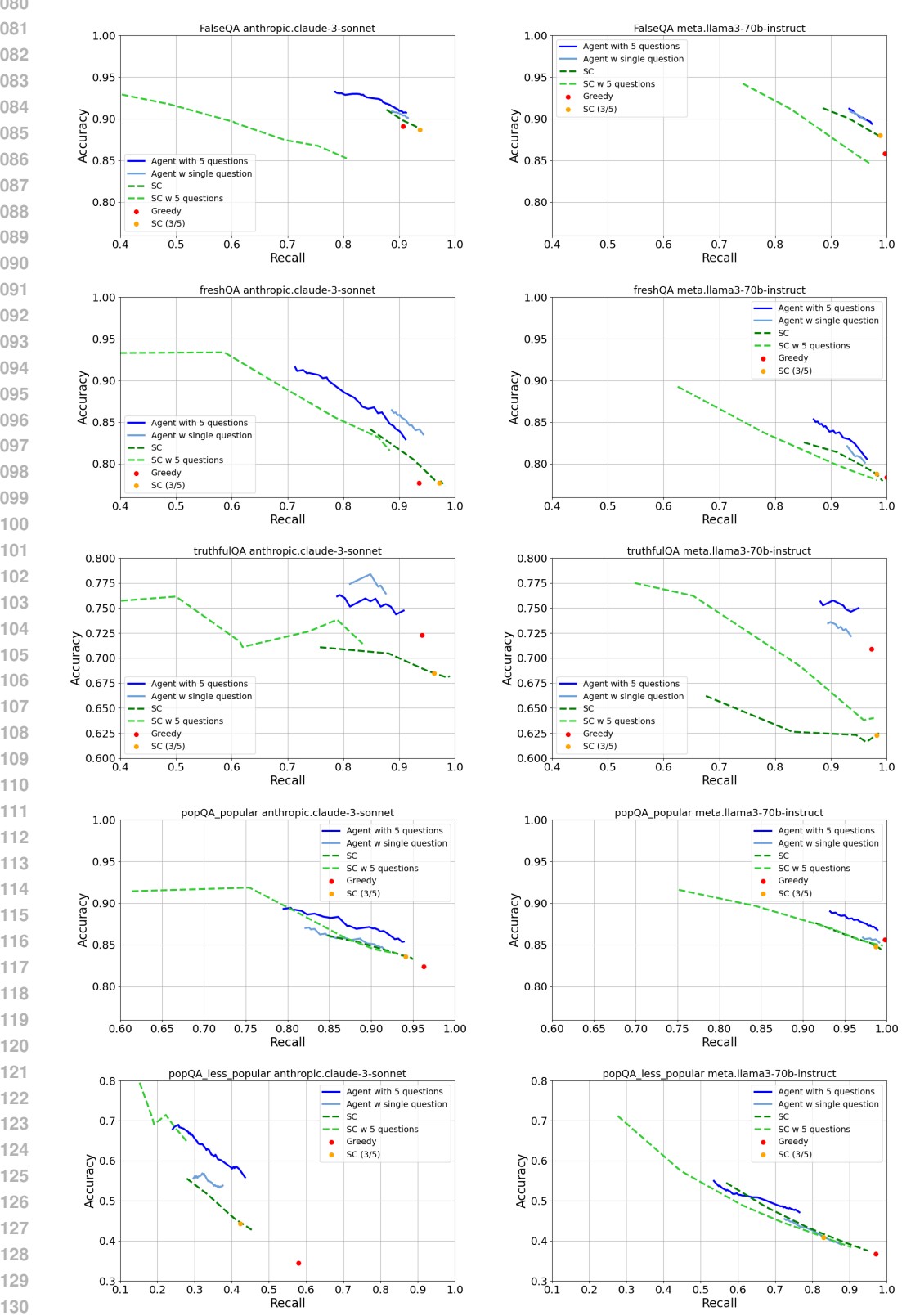

Figure 8: AR-curves for both the baselines and the proposed method on individual datasets. SC refers to self-consistency-based entropy. SC w 5 questions refers to the baseline using the agents' questions without agent interaction.

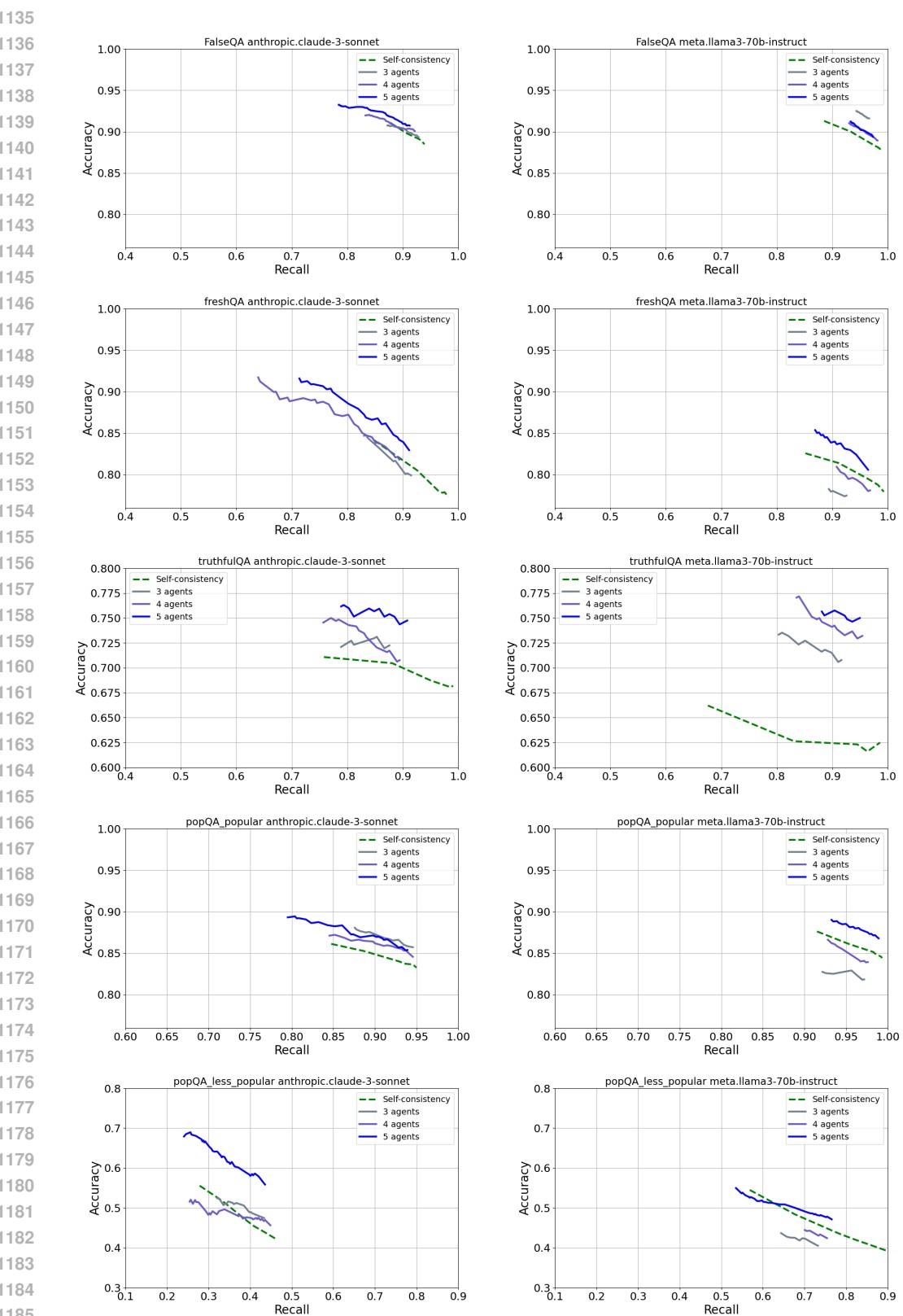

Figure 9: We present the effect of the number of agents on agent performance for each dataset.

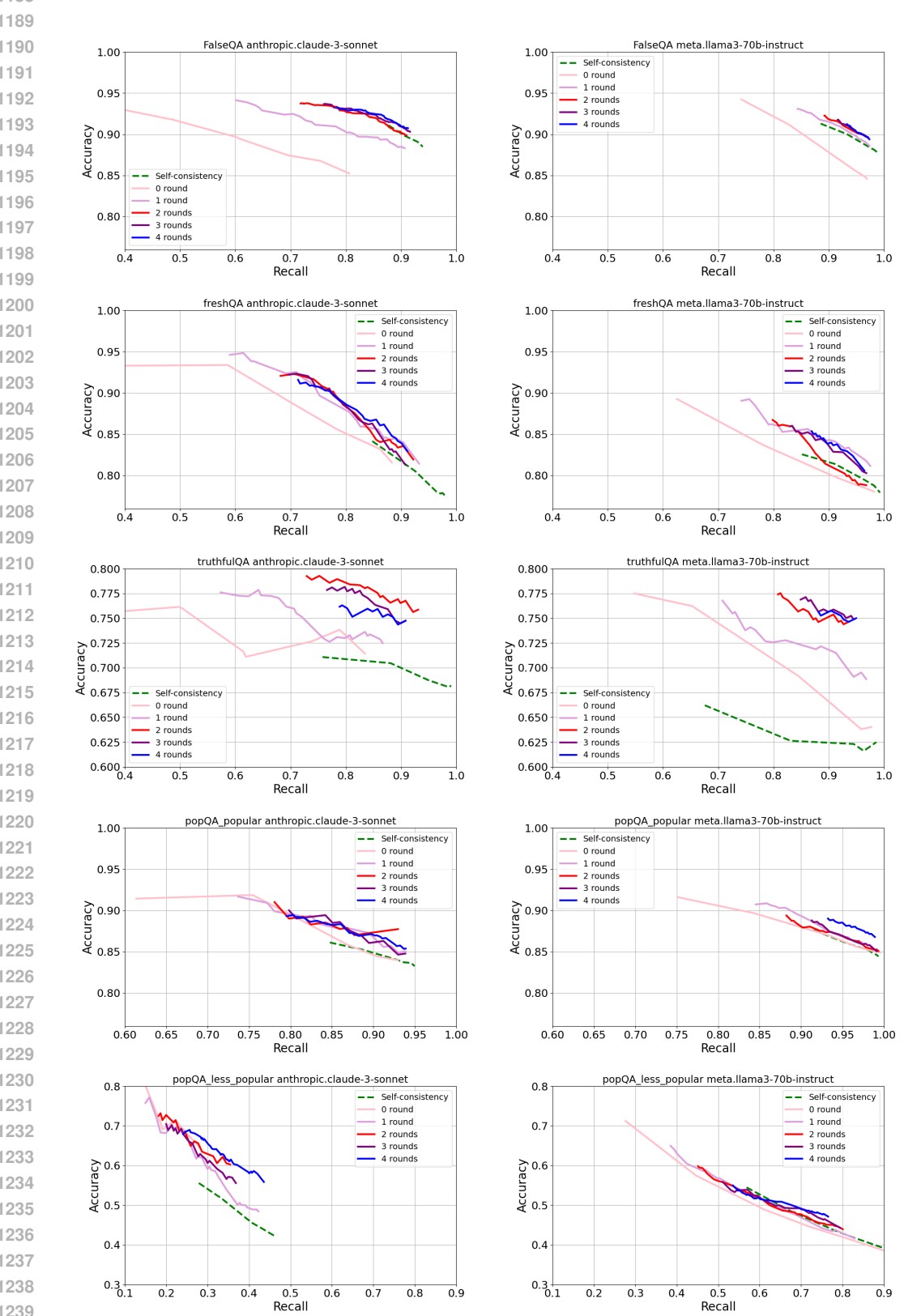

Figure 10: We present the effect of the number of interactions on agent performance on each dataset.

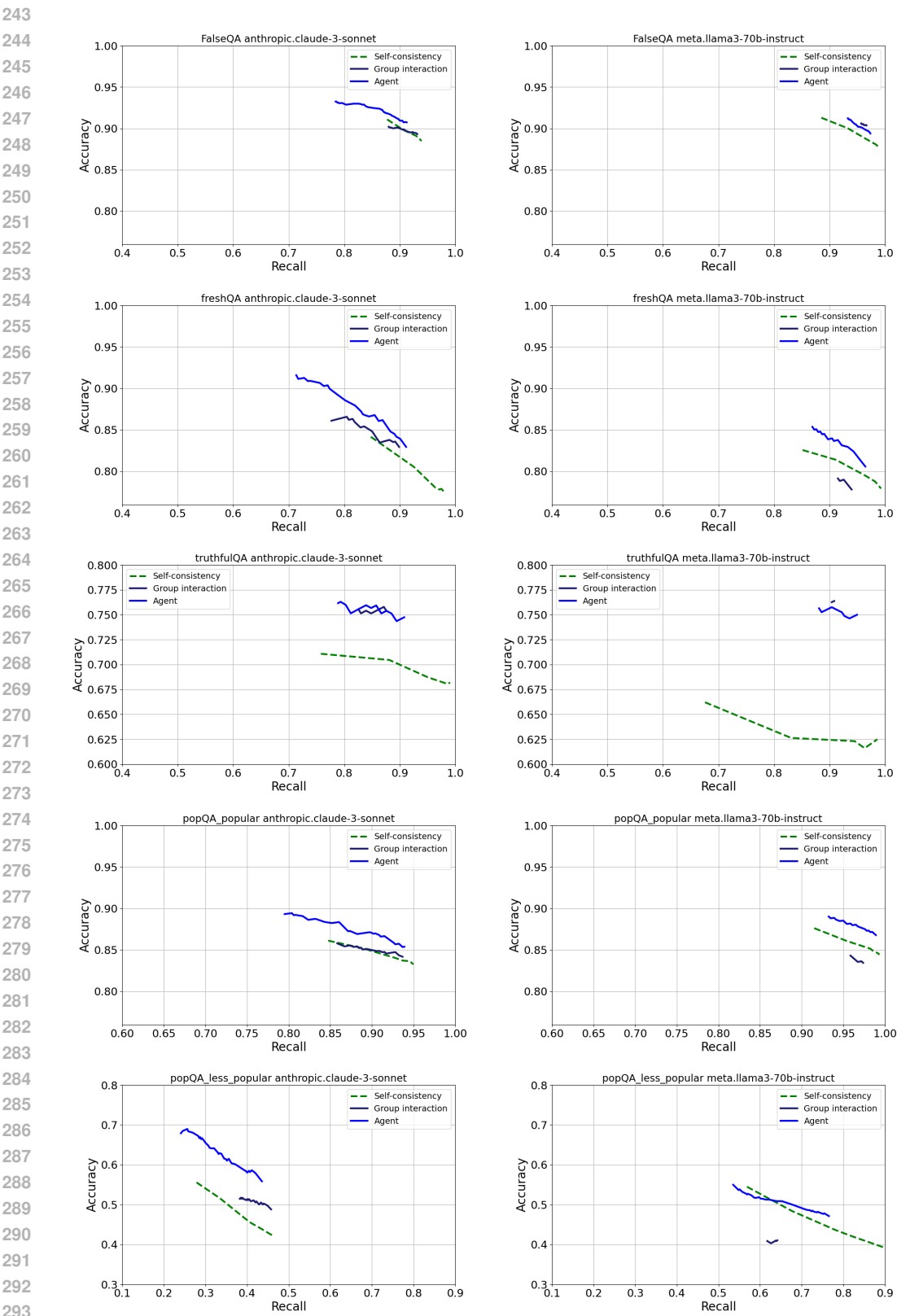

Figure 11: We present the effect of interaction format on each dataset.

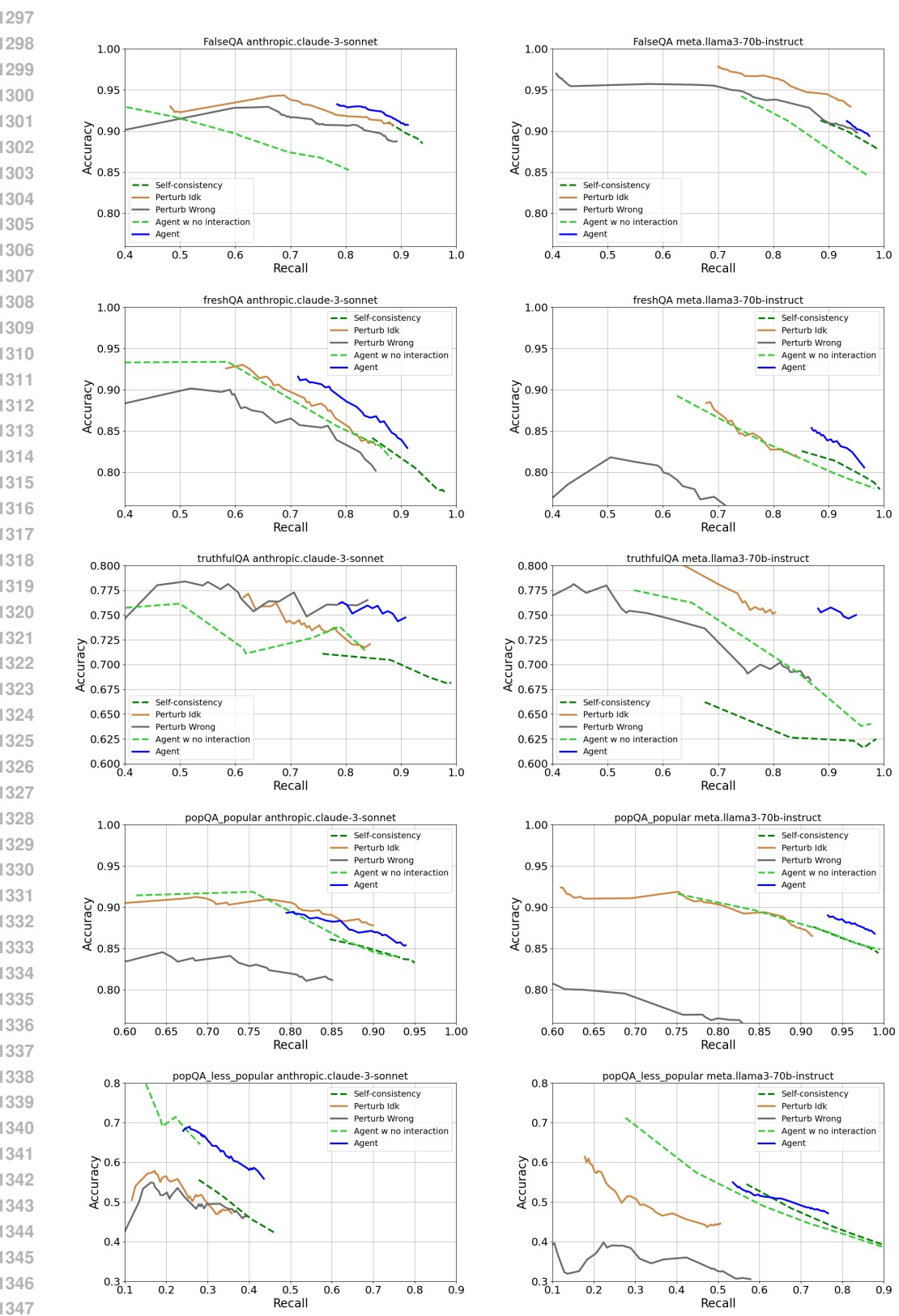

Figure 12: We present the robustness of interaction on each dataset.

## A.10 PROMPTS FOR THE PROPOSED DIVERSEAGENTENTROPY METHOD

---

**EXAMPLE QUESTION CONCEPTUALIZTION PROMPT**

**SYSTEM**
Can you identify the broader category of the specific entity
referenced in the question?  If there is a specific entity, you
MUST CHANGE it to a general category, e.g., a person, a item, a
place, a object.  If there is no specific entity, you MUST KEEP the
original question.
**USER**
What is the most spoken language in the world?
**ASSISTANT**
What is the most spoken language in the world?
**USER**
What is a Joe Biden's occupation?
**ASSISTANT**
What is a person's occupation?

Figure 13: Example **Question Conceptualization** Prompt

---

**EXAMPLE ASPECT GENERATION PROMPT**

**SYSTEM**
Can you identify up to 5 key conceptual perspectives that are
as varied and diverse as possible, ensuring a comprehensive
and multifaceted understanding of the question?  Given ONLY the
conceptual aspect name, no other words or explanation.  The aspect
SHOUlD NOT indicate the answer to the question.  Each aspect is a
line <as short as possible; not a complete sentence!>
**USER**
What is the most spoken language in the world?
**ASSISTANT**
demographic statistics
education policy
cultural influence
technology and media
globalization effects

Figure 14: Example **Aspect Generation** Prompt

---

**EXAMPLE ASPECT QUESTION GENERATION PROMPT**

**SYSTEM**
Generate 5 questions that build upon a given question based on a given aspect. Ensure that each question STRICTLY requires knowledge of this original question to answer but DO NOT INCLUDE the direct answer to the original question in your generated questions but MUST INCLUDE the exact content of the original question. These questions should encourage a deeper exploration of the underlying themes or concepts introduced in the original question. Given ONLY the question, no other words or explanation. For example: Q1: <the generated question should be SIMPLE and DO NOT INCLUDE the direct answer to the original question>

**USER**
Question: What is the most spoken language in the world?
Aspect: cultural influence

**ASSISTANT**
Q1: How does the prevalence of the most spoken language in the world influence global media and entertainment?
Q2: In what ways has the most spoken language in the world shaped international business and trade practices?
Q3: How has the education system in non-native speaking countries adapted to teach the most spoken language in the world?
Q4: What role does the most spoken language in the world play in diplomatic relations and international policy-making?
Q5: How has the cultural heritage of the most spoken language in the world influenced global cuisine and fashion trends?

Figure 15: Example **Aspect Question Generation** Prompt

---

**EXAMPLE SEMANTICALLY EQUIVALENT QUESTION GENERATION PROMPT**

**SYSTEM**
For the given question, provide 5 semantically equivalent questions. Do not answer the question. STRICTLY follow the structure that each generated question is a line.

**USER**
What is the most spoken language in the world?

**ASSISTANT**
Which language has the highest number of speakers globally?
What language is spoken by most people worldwide?
Which language tops the list of the world's most widely spoken languages?
What is the world's dominant language by number of speakers?
Globally, which language is spoken by the greatest number of people?

Figure 16: Example **Semantically Equivalent Question Generation** Prompt

---

**EXAMPLE 1-1 INTERACTION PROMPT**

**SYSTEM**
You are an AI assistant that helps people answer questions. Ensure your responses are concise and strictly relevant to the queries presented, avoiding any unrelated content to the question. Do not change your answer unless you think you are absolutely wrong.
<previous interaction conversations...>
**USER**
When I asked you in another api call that + selection_agent_question + You mentioned that + selection_agent_answer_to_original_query + Which is your actual answer to + original_query?

Figure 17: Example **1-1 Interaction** Prompt

---

**EXAMPLE ANSWER EXTRACTION PROMPT**

**SYSTEM**
You will extract the answer to the given question using ONLY the information provided in the "Response" section. You will identify the answer directly without using any additional knowledge or explanation. If the response includes a negation to the question, use those as the answer.
**USER**
Response: The prevalence of the most spoken language in the world, which is Mandarin Chinese, has a significant influence on global media and entertainment in several ways:
1. **Content creation**: Many Chinese production companies and studios create content specifically for the massive Chinese-speaking audience, which often gets distributed globally. This leads to a increase in Chinese-language content in international markets. 2. **Dubbing and subtitles**: To cater to the large Chinese-speaking population, many international films and TV shows are dubbed or subtitled in Mandarin, making them more accessible to Chinese audiences. 3. **Global market appeal**: The massive Chinese market has become a crucial factor in the success of global films, TV shows, and music. Creators often tailor their content to appeal to Chinese audiences, incorporating Chinese themes, actors, or storylines. 4. **Influence on global trends**: Chinese social media platforms, such as WeChat and Weibo, have become essential channels for promoting global entertainment content. Trends and memes that originate in China can quickly spread globally. 5. **Cultural exchange**: The prevalence of Mandarin Chinese has facilitated cultural exchange between China and other countries. International artists and creators are increasingly collaborating with Chinese counterparts, leading to a fusion of cultural influences in media and entertainment.
Overall, the dominance of Mandarin Chinese has reshaped the global media and entertainment landscape, with creators and distributors adapting to cater to the vast and influential Chinese-speaking audience.
Based solely on the response, What is the most spoken language in the world?
**ASSISTANT**
The most spoken language in the world is Mandarin Chinese.

Figure 18: Example **Answer Extraction** Prompt

