# OpenReview forum: "DiverseAgentEntropy: Quantifying Black-Box LLM Uncertainty through Diverse Perspectives and Multi-Agent Interaction"
_ICLR.cc/2025/Conference — Submitted to ICLR 2025_

### Official Review · Reviewer_aMZi · 2024-10-27

**Soundness:** 3
**Presentation:** 3
**Contribution:** 3
**Rating:** 6
**Confidence:** 4

**Summary:**

This paper introduces DIVERSEAGENTENTROPY, a novel method for quantifying uncertainty in Large Language Models (LLMs) through multi-agent interaction. Unlike traditional self-consistency methods that rely on repeated sampling of the same query, this approach generates diverse questions about the same underlying fact and creates multiple "agents" from the same base model to answer these questions. The method showed a 2.5% improvement in accuracy compared to existing self-consistency approaches and revealed important insights about LLMs' ability to consistently retrieve knowledge across different contexts.

**Strengths:**

1, This paper introduces a more comprehensive approach to uncertainty estimation that goes beyond simple self-consistency
2, the experiments are across multiple datasets and model types. The results are promising

**Weaknesses:**

1, Requires significantly more computational resources (5x more API calls) compared to traditional methods
2, it is primarily tested on factual question-answering tasks and may not generalize well to other types of language tasks

**Questions:**

NA

---

> ### Author Response · Authors · 2024-11-21
>
> We sincerely thank reviewer aMZi for the constructive suggestions.  Here we clarify the questions you posed.
>
> **Computational cost**
>
> We acknowledge that the cost of our method is relatively higher compared to self-consistency-based methods. However, we emphasize the following points:
>
> 1. Superior performance: our method outperforms simple self-consistency-based approaches. In high-stakes applications where correctness is prioritized over cost, our calibrated uncertainty score provides users with a reliable measure of how much they can trust the model's output. Additionally, the chosen answers after applying the abstention policy are more accurate.
>
> 2. Utility for finetuning: the intermediate results generated by our method, including varied questions and the self-reflection interaction processes, can be further leveraged to create synthetic data for finetuning or training LLMs.
>
> 3. Potential for optimization: future work can explore ways to maintain the same level of performance while reducing costs. This could involve using fewer but higher-quality questions from diverse perspectives and minimizing the number of interaction rounds.
>
> We have added a detailed analysis to present the cost/number of inference calls for each method in Appendix Table 6 in the updated version.
>
> **Generalization to other types of language tasks**
>
> We believe our method has the potential to generalize to a wide range of QA tasks, including those involving RAG and complex QA scenarios. Its ability to generate questions from diverse perspectives and leverage multi-agent interactions makes it well-suited for handling complex reasoning and integrating diverse information sources effectively. For RAG, by generating questions from different perspectives, we prompt the retriever to fetch diverse documents, enriching the range of potential information sources. Additionally, the multi-agent interaction facilitates self-reflection, which is particularly beneficial for QA scenarios involving conflicting retrieved documents. For complex QA, in order to effectively use our pipeline, we propose future work to incorporate an additional meta judge to monitor agents' overall understanding of the query. This mechanism ensures that the model does not resort to taking shortcuts, such as prematurely considering the query invalid.

---

### Official Review · Reviewer_YVnz · 2024-11-01

**Soundness:** 4
**Presentation:** 4
**Contribution:** 4
**Rating:** 6
**Confidence:** 4

**Summary:**

This paper presents a novel method, named DIVERSEAGENTENTROPY, to quantify the uncertainty of LLMs. The assumption is that: if the model is certain of its answer to a query, it should consistently provide the same answer across variants of the original query. In the proposed method, given a new query, varies questions are generated using LLMs. Each agent will answer a unique varied question independently, then interact with another agent in a controlled cross-play one-on-one manner. The final uncertainty is calculated to obtain the final answer. Extensive experiments are conducted to test the performance of the proposed method.

**Strengths:**

* Overall, the paper is very clear and easy to follow.
* The paper focuses on an emerging problem in the era of LLMs. In the proposed method, the LLM is viewed as a black-box. Therefore, the proposed method can be easily applied to other LLMs.
* The paper is well-motivated. The examples in Fig. 1 describe the problem well.
* The experiments are very comprehensive to validate the performance of the proposed method.

**Weaknesses:**

My main concern is the extra LLM calls in the proposed framework, which may limit the practicality of the proposed method. As discussed in Appendix A.1, the number of LLM calls is much higher than the self-consistency-based approaches. In addition, there are dependencies among the tasks in round 1 -- independent QA and round 2 -- 1-1 interaction, and each round of interaction in round 2, which will make the latency cost higher.

**Questions:**

1. When generating the varied questions, how can we ensure the answer of the generated question is the same as the original query?
2. In line 223, how is the answer extracted from the response?
3. In line 231, how is the difference between the answers measured? Since LLM may output a full answer (as shown in Table 8), or add unnecessary information in the response, will an answer extraction be applied after each 1-1 interaction?
4. To make the round 2 1 - 1 interaction (as described in line 233-234) more clear, will it be good to share the prompt template?
5. This work focuses on the factual parametric knowledge of LLMs, are there any insights on whether the method can be applied to RAG?

---

> ### Author Response · Authors · 2024-11-21
>
> We sincerely thank reviewer YVnz for the constructive suggestion.  Here we clarify the questions you posed.
>
> **Computational cost**
>
> We acknowledge that the cost of our method is relatively higher compared to self-consistency-based methods. However, we emphasize the following points:
>
> 1. Superior performance: our method outperforms simple self-consistency-based approaches. In high-stakes applications where correctness is prioritized over cost, our calibrated uncertainty score can provide users with a reliable measure of how much they can trust the model's output. Additionally, the chosen answers after applying the abstention policy are more accurate.
>
> 2. Utility for finetuning: the intermediate results generated by our method, including varied questions and the self-reflection interaction processes, can be further leveraged to create synthetic data for finetuning or training LLMs.
>
> 3. Potential for optimization: future work can explore ways to maintain the same level of performance while reducing costs. This could involve using fewer but higher-quality questions from diverse perspectives and minimizing the number of interaction rounds.
>
> We have added a detailed analysis to present the cost/number of inference calls for each method in Appendix Table 6 in the updated version.
>
> **Questions about the implementation of the pipeline**
>
> We ensure that the original query is preserved in the context of the generated varied questions, but not necessarily the original answer. After generating these varied questions, we immediately prompt the model to self-check whether each generated question strictly requires knowledge of the original query to answer. We adopt the following prompt for extracting the answer to the original query from the response. An answer to the original query will also be extracted after each 1-1 interaction.
>
> ```
> System: You will extract the answer to the given question using ONLY the information provided in the "Response" section. You will identify the answer directly without using any additional knowledge or explanation. If the response includes a negation to the question, use those as the answer.
>
> User: Response: <response>
> Assistant: <extracted answer>
> ```
> We show the prompt for 1-1 interaction below.
>
> ```
> System: You are an AI assistant that helps people answer questions. Ensure your responses are concise and strictly relevant to the queries presented, avoiding any unrelated content to the question. Do not change your answer unless you think you are absolutely wrong.
> <previous interaction conversations…>
> User: “When I asked you in another API call that” + selection_agent_question + “You mentioned that’’+ selection_agent_answer_to_original_query + “Which is your actual answer to” + original_query?
> Assistant:  <generate a new answer to the original query>
> ```
> Note that we have added both prompts to the paper in line 297 and the appendix prompts.
>
> **Application to RAG**
>
> Our method can be applied to tasks like RAG. By generating questions from different perspectives, we can prompt the retriever to fetch diverse documents, enriching the range of potential information sources. Additionally, the multi-agent interaction facilitates self-reflection, which is particularly beneficial for QA scenarios involving conflicting retrieved documents.

---

### Official Review · Reviewer_D3Hu · 2024-11-02

**Soundness:** 2
**Presentation:** 3
**Contribution:** 2
**Rating:** 3
**Confidence:** 4

**Summary:**

This paper proposes a new method for estimating the uncertainty of language model outputs.
There are two core steps:
1. Prompt the model to answer a set of equivalent but diverse questions from different perspectives.
2. Allow agent interaction by prompting the LLM to reconcile each pair of different answers from the first step.

The uncertainty of each semantically unique answer is then their weighted frequency, where the weightage is how frequently the agents changes its answer during the second step of agent interaction.

The experiments are done on two models (Claude-3-Sonnet and Llama-3-70b-Instruct) on several QA datasets, and the metrics focus on selective prediction performance. Results show that the proposed method is better than the Self-Consistency baseline which simply takes majority vote the multiple sampled solutions.

My biggest concern about this work is its limited novelty. The proposed method feels similar to existing multi-agent debate work, combined with the sef-consistency and semantic entropy ideas (which are not new either). The authors should better highlight what's the novelty contribution of their proposed framework as compared to existing works.

**Strengths:**

- The paper is generally well-organized and clearly written.
- The experiments are reasonably thorough, with comparisons to many relevant baselines.

**Weaknesses:**

1. Limited novelty.
It seems every step of the proposed framework is not new.
For generating paraphrased/equivalent questions and checking answer consistency, there is "SAC3: Reliable Hallucination Detection in Black-Box Language Models via Semantic-aware Cross-check Consistency" (EMNLP'23). For multi-agent debate, there is "Improving Factuality and Reasoning in Language Models through Multiagent Debate" (ICML 2024). For aggregating diverse answers into uncertainty scores, there is "Semantic Uncertainty: Linguistic Invariances for Uncertainty Estimation in Natural Language Generation" (ICLR 2023). So it seems to me that the proposed framework is a combination of known techniques for the calibration problem. This feels too incremental for the ICLR standard.

2. Some nitpick about how you reported the results. I think using AUC as the main metric is fine since you mostly target calibration / selective prediction. But I think you should include all the baselines in Table 1 to contextualize the results of your proposed method. Ideally you might also include a column for cost / number of inference calls. The different metrics in Table 2 are starting to get a bit confusing to me. Why not still use AUC or metrics like Cov@Acc? (E.g., see "Selective Question Answering under Domain Shift" ACL 2020).

**Questions:**

N/A

---

> ### Author Response · Authors · 2024-11-21
>
> We sincerely thank reviewer D3Hu for the valuable comments. Here we answer all the questions and hope they can address your concerns.
>
> **Novelty**
>
> We would like to highlight two major novelties for our method:
>
> 1. Our work introduces **diverse perspective** questions by injecting additional context from different perspectives into the original query. For example, in Figure 1, "What is the most common symptom experienced by women with the leading cause of cancer deaths in the U.S.?" adds context about the cancer diagnosis perspective to the original query "What type of cancer kills the most women in the U.S.? ". This approach differs from works like Semantic Uncertainty (ICLR 2023), which focuses on the original query, and SAC3 (EMNLP 2023), which only paraphrases queries into semantically equivalent forms without adding explicit new context. In fact, both tested models in our paper will still consistently give the wrong answer to the semantically equivalent form of the original query, e.g., “Which organ does the cancer that kills the most US women affect?” . These diverse perspective questions further enable unique agent backgrounds for same-model multi-agent interactions (lines 220-224) and facilitate a novel and unique analysis of the retrievability of a model, showing that models fail to consistently retrieve the correct answer to the same query under diverse perspective questions (lines 416-454).
>
> 2. Unlike the debate framework in Improving Factuality and Reasoning in Language Models through Multiagent Debate (ICML 2024), which involves agents from different models debating and persuading each other to improve downstream performance, our approach focuses on same-model agent interaction. By exposing the same model to different contextual hints, we aim to analyze an individual model’s uncertainty more effectively. In this setting, we allow the model to state "I don’t know" rather than forcing it to generate an answer, emphasizing a fundamentally different goal and interaction paradigm. We also propose a novel weighted algorithm combined with classical entropy for uncertainty estimation during agent interactions (lines 255-269). To the best of our knowledge, we are the first to develop a method for measuring a model’s uncertainty after agent interaction.
>
> **Evaluation metrics**
>
> We address both uncertainty estimation and hallucination detection in our work. In Table 1, we compare our approach with rigorous uncertainty estimation methods to evaluate the calibration of our method ( We adopt three new popular and recent uncertainty estimation baselines from Generating with Confidence: Uncertainty Quantification for Black-box Large Language Models (TMLR 2024), see lines 196-202). In Table 2, we benchmark our approach against hallucination detection and direct answering methods to assess its ability to identify and mitigate hallucinations effectively.
>
> We evaluate AUROC in Table 1, as it is the primary metric used in uncertainty estimation papers for assessing calibration and ranking ability in a threshold-independent manner. However, AUROC is less informative for final question-answering scenarios, where users expect a model for either a single correct answer or an explicit acknowledgment of uncertainty, such as stating, "I don’t know."
>
> To address this, we employ the evaluation metrics shown in Table 2, which focus on the final question answering with the optimal threshold that a user would select for each method. Our results demonstrate that our method achieves the highest accuracy on questions where the model does not abstain from answering. Additionally, it achieves the highest correctness and informativeness across the entire dataset, metrics that are also used in TruthfulQA (ACL 2022). This indicates that for users seeking an optimal answer for each question, our proposed method is the best choice.
>
> Furthermore, as presented in Figure 3, we evaluate cov@acc by sampling all possible coverage rates and plotting their corresponding accuracy. The results confirm that, among coverage rates where all methods are applicable, our method consistently achieves the highest accuracy.
>
> We agree with the reviewer that we should also present the cost/number of inference calls for each method. We have added a detailed analysis in Appendix Table 6 in the updated version.

---

> > ### Comment · Reviewer_D3Hu · 2024-11-24
> > **update**
> >
> > Thanks for the update. I unfortunately can't be convinced by the novelty justifications and so will keep my score.

---

> > > ### Author Response · Authors · 2024-11-26
> > >
> > > Thank you for your additional comment! Since we have been granted extra rebuttal time, could you kindly elaborate on why you remain unconvinced by our novelty justifications? We have compared our work to the related works you mentioned and explained in detail how our approach differs and incorporates innovative methods. We would greatly appreciate the opportunity to address any specific concerns you may have further.

---

### Official Review · Reviewer_KJxh · 2024-11-06

**Soundness:** 3
**Presentation:** 4
**Contribution:** 3
**Rating:** 3
**Confidence:** 4

**Summary:**

In this paper, the authors propose a novel method for LLM's uncertainty estimation, which quantifies the LLMs' uncertainty based on the consistency of responses across diverse questions after multi-agent interaction.

**Strengths:**

1. This paper is well-written. The structure is clear and easy to follow.

2. The idea of measuring UE of LLM via different perspective from different agents is clear and make sense.

3. The design of method is also clear and reasonable.

**Weaknesses:**

1. Achieve UE by considering the multiple results from various aspect (multiple agents in this paper) is not very new and novel compared to some recent papers: Knowing What LLMs DO NOT Know: A Simple Yet Effective Self-Detection Method（NAACL 2024）, SelfCheckGPT: Zero-Resource Black-Box Hallucination Detection for Generative Large Language Models. (EMNLP 2023)

2. It would be more solid to compare with some more recent baselines. The baselines are mainly in 2023 or eailer. SC is a very basic baseline model. As uncertainty estimation (UE) is becoming hot recently, there should be some advanced baselines, such as "Know-
ing What LLMs DO NOT Know: A Simple Yet Effective Self-Detection Method". In NAACL

3. From the experimental results shown in Tables 1 & 2, the proposed model cannot always excel the performance of baselines (In table 1 & 2).

4. The related work section should be enriched to provide a more comprehensive overview of related research.

**Questions:**

None

---

> ### Author Response · Authors · 2024-11-21
>
> We sincerely thank reviewer KJxh for the valuable comments. Here we answer all the questions and hope they can address your concerns.
>
> **Novelty**
>
> We would like to highlight two major novelties for our method:
>
> 1. Our work introduces **diverse perspective** questions by injecting additional context from different perspectives into the original query. For example, in Figure 1, "What is the most common symptom experienced by women with the leading cause of cancer deaths in the U.S.?" adds context about the cancer diagnosis perspective to the original query "What type of cancer kills the most women in the U.S.? ". This approach differs from works like SelfCheckGPT (EMNLP 2023), which focus on the original query, and Knowing What LLMs Do Not Know (NAACL 2024) or SAC3 (EMNLP 2023), which only paraphrase queries into semantically equivalent forms without adding explicit new context. In fact, both tested models in our paper will still consistently give the wrong answer to the semantically equivalent form of the original query, e.g., “Which organ does the cancer that kills the most US women affect?” . These diverse perspective questions further enable unique agent backgrounds for same-model multi-agent interactions (lines 220-224) and facilitate a novel and unique analysis of the retrievability of a model, showing that models fail to consistently retrieve the correct answer to the same query under diverse perspective questions (lines 416-454).
>
> 2. None of the mentioned methods introduce **multi-agent interaction**, which is crucial in our approach to enable the model to be exposed to diverse perspectives and self-reflect. We propose a weighted algorithm combined with classical entropy for uncertainty estimation during agent interactions(lines 255-269). To the best of our knowledge, we are the first to develop a method for measuring a model’s uncertainty after agent interaction.
>
> **Baselines**
>
> We would like to emphasize that our primary goal is to compare our method with entropy-based uncertainty estimation methods, as these are the most commonly used approaches in the literature for uncertainty estimation in NLG. ( We already highlighted this in Section 3.1 and have revised Section 3.2 to clarify this focus). Specifically, we compare against Self-Consistency(SE), also known as SemanticEntropy, which is a widely recognized and effective uncertainty estimation method (ICLR 2023, Nature 2024). Additionally, we adopt three new and popular black-box uncertainty estimation baselines from Generating with Confidence: Uncertainty Quantification for Black-box Large Language Models (TMLR 2024). We have added a description of these methods in lines 196–202 and present their results in Table 1.
>
> We acknowledge that Knowing What LLMs Do Not Know (NAACL 2024) is a very related paper. However, it is proposed as a nonfactuality detection method, not an uncertainty estimation method. We will thus use it as a hallucination detection method for comparison in Table 2. Since the codebase of this paper lacks sufficient details for proper implementation of the Atypicality component and XGBoost fitting, we emailed the authors for clarification and plan to include the full baseline in the final version. Nonetheless, SeQ in Table 2 is a simple version of Knowing What LLMs Do Not Know (NAACL 2024) without the Atypicality component.
>
> Tables 1&2 and Figure 3 show that our method consistently performs better than any baselines in terms of uncertainty score calibration (Table 1) and hallucination detection (Table 2 and Figure 3).  We highlight that ab-R is not our primary metric, as it only reflects how often the model abstains and closely correlates with accuracy (acc). With similar ab-R values, the method with higher accuracy (acc) is preferable.
>
> **Related works**
>
> We agree with the reviewer that related work should be enriched and we have added more related papers including the ones that the reviewer has mentioned in the related work section.

---

### Meta-Review · Area_Chair_2p7M · 2024-12-20

**Metareview:**

This paper introduces DiverseAgentEntropy, a method to evaluate uncertainty in Large Language Models (LLMs) using multi-agent interactions in a black-box setting, claiming enhanced reliability and hallucination detection capabilities. The paper is structured clearly and demonstrates detailed experimental work. However, as pointed out by the reviewers, there are several concerns regarding the novelty of the approach, particularly its similarity to existing methods that also use multi-agent interactions and semantic checks for uncertainty estimation. Moreover, the experimental setup's dependence on specific task types raises questions about the generalizability of the method across different LLM applications, and the method's efficiency in terms of computational resources and response latency needs further justification considering its scalability in real-world scenarios. Even though the authors attempted to address these points during the rebuttal, the reviewers were not fully satisfied with the answers.

**Additional Comments On Reviewer Discussion:**

Nil.

---

### Decision · Program_Chairs · 2025-01-22

Reject